# “It Empowers You to Empower Them”: Health Professional Perspectives of Care for Hyperglycaemia in Pregnancy Following a Multi-Component Health Systems Intervention

**DOI:** 10.3390/ijerph21091139

**Published:** 2024-08-28

**Authors:** Diana MacKay, Louise Maple-Brown, Natasha Freeman, Jacqueline A. Boyle, Sandra Campbell, Anna McLean, Sumaria Corpus, Cherie Whitbread, Paula Van Dokkum, Christine Connors, Elizabeth Moore, Ashim Sinha, Yvonne Cadet-James, John Boffa, Sian Graham, Jeremy Oats, Alex Brown, H. David McIntyre, Renae Kirkham

**Affiliations:** 1Menzies School of Health Research, Charles Darwin University, Darwin 0810, Australia; 2Department of Endocrinology, Royal Darwin Hospital, Darwin 0810, Australia; 3Eastern Health Clinical School, Monash University, Melbourne 3128, Australia; 4Jawun Research Centre, Central Queensland University, Cairns 4870, Australia; 5Department of Diabetes and Endocrinology, Cairns and Hinterland Hospital and Health Service, Cairns 4870, Australia; 6Department of Diabetes, Royal Darwin Hospital, Darwin 0810, Australia; 7Danila Dilba Health Service, Darwin 0800, Australia; 8Alice Springs Hospital, Alice Springs 0870, Australia; 9Northern Territory Department of Health, Darwin 0800, Australia; 10Aboriginal Medical Services Alliance Northern Territory, Darwin 0800, Australia; liz.moore@amsant.org.au; 11Apunipima Cape York Health Council, Bungalow 4870, Australia; 12Central Australian Aboriginal Congress, Alice Springs 0800, Australia; 13Melbourne School of Population and Global Health, University of Melbourne, Melbourne 3053, Australia; 14College of Health and Medicine, Australian National University, Canberra 2601, Australia; 15Telethon Kids Institute, Perth 6009, Australia; 16Mater Research, The University of Queensland, Brisbane 4101, Australia

**Keywords:** diabetes in pregnancy, gestational diabetes, health systems, quality improvement, Aboriginal health, First Nations, mixed methods

## Abstract

The Northern Territory (NT) and Far North Queensland (FNQ) have a high proportion of Aboriginal and Torres Strait Islander women birthing who experience hyperglycaemia in pregnancy. A multi-component health systems intervention to improve antenatal and postpartum care in these regions for women with hyperglycaemia in pregnancy was implemented between 2016 and 2019. We explored health professional perspectives on the impact of the intervention on healthcare. The RE-AIM framework (Reach, Effectiveness, Adoption, Implementation, Maintenance) underpinned this mixed-methods evaluation. Clinicians were surveyed before (*n* = 183) and following (*n* = 137) implementation. The constructs explored included usual practice and satisfaction with care pathways and communication between services. Clinicians, policymakers and the implementation team were interviewed (*n* = 36), exploring the impact of the health systems intervention on practice and systems of care. Survey and interview participants reported improvements in clinical practice and systems of care. Self-reported glucose screening practices improved, including the use of recommended tests (72.0% using recommended first-trimester screening test at baseline, 94.8% post-intervention, *p* < 0.001) and the timing of postpartum diabetes screening (28.3% screening at appropriate interval after gestational diabetes at baseline, 66.7% post-intervention, *p* < 0.001). Health professionals reported multiple improvements to care for women with hyperglycaemia in pregnancy following the health systems intervention.

## 1. Introduction

In countries with a history of colonisation, First Nations people experience a disproportionate burden of diabetes [1,2]. In the face of colonisation, Australia’s First Nations peoples—Aboriginal and Torres Strait Islander people—have demonstrated great resiliency, maintaining an enduring connection to country, language and cultural practices, which has lasted for over 60,000 years [3]. Despite these strengths, Aboriginal and Torres Strait Islander women are more likely than non-Indigenous Australian women to be diagnosed with hyperglycaemia in pregnancy [4,5], which encompasses gestational diabetes (GDM), diabetes in pregnancy (DIP, generally defined as newly-detected hyperglycaemia which meets criteria for diagnosis of diabetes outside of pregnancy) and pre-existing diabetes. Hyperglycaemia in pregnancy confers an increased risk of pregnancy and birth complications, such as stillbirth, pre-eclampsia and macrosomia [6,7]. Additionally, there are long-term risks to both mother and child, including increased risk of diabetes for women who have had GDM [8] and increased risk of overweight and type 2 diabetes (T2D), particularly onset at a young age, for the infant [9]. Hyperglycaemia in pregnancy is thus an important contributor to the high prevalence of diabetes among First Nations people.

Antenatal and postpartum care for women with hyperglycaemia in pregnancy often falls short of recommended standards, with the International Federation of Gynecology and Obstetrics recognising improving systems of care for hyperglycaemia in pregnancy as a research priority [10]. Several studies have demonstrated improvements in care during and after pregnancy through health system changes [11,12,13]. However, previous studies have included only women with either GDM or pre-existing diabetes, rather than acknowledging the need to improve care for women across the spectrum of glucose intolerance in pregnancy. Additionally, there is an absence of reported programs that have aimed to specifically improve care for First Nations women. 

There are distinct challenges to providing optimal healthcare for women with hyperglycaemia in pregnancy in the regional and remote setting. In Australia’s Northern Territory (NT) and Far North Queensland (FNQ), previously identified barriers to care provision included disjointed communication and care coordination, lack of role clarity between care providers, high turnover of clinical staff, limited workforce capacity (especially of Aboriginal and Torres Strait Islander staff) and challenges in communicating effectively across cultural and linguistic barriers [14,15,16,17,18]. 

The Diabetes Across the Lifecourse: Northern Australia Partnership (“the Partnership”), was formed in the NT in 2012 as a collaboration between clinicians, health services, researchers and policymakers. The Partnership expanded to include FNQ in 2015. A multi-component health systems intervention was implemented by the Partnership between 2016 and 2020, which aimed to improve antenatal and postpartum care for women with hyperglycaemia in pregnancy in the NT and FNQ. Here we report health professionals’ perceptions of care provision during and after a pregnancy complicated by hyperglycaemia, following the implementation of the health systems intervention.

## 2. Methods

### 2.1. Setting

The NT and FNQ has a small population of approximately 500,000, dispersed over a vast geographic area of 1.6 million km^2^, including numerous small islands and other remote communities which become inaccessible by road during the monsoonal wet season [19,20,21]. Among NT residents, 27%, and in FNQ 19%, identify as Aboriginal and/or Torres Strait Islander people, compared with 3.2% nationally [19,20,21,22]. Over 200 languages are spoken across the regions, including Aboriginal and Torres Strait Islander languages, as well as those of migrant populations [19,20,21]. Annually, across the NT and FNQ there are approximately 7000 births per year, hyperglycaemia complicating 18.6% of pregnancies in the NT and 20% of pregnancies in FNQ [23,24]. In the NT in 2019, 14.5% of pregnancies to First Nations women were complicated by GDM and 4.9% by pre-existing diabetes [24]. 

A complex network of hospital and primary care services is involved in the provision of care for women with hyperglycaemia in pregnancy. Primary care is provided by either an Aboriginal Community Controlled Health Service, a state/territory government clinic or a private provider. The multidisciplinary care required for women with hyperglycaemia in pregnancy usually involves care providers across multiple services.

### 2.2. Health Systems Intervention Design

In a collaboration between researchers, clinicians, health services and policymakers, we developed a multi-component health systems intervention to improve antenatal and postpartum care for women in the NT and FNQ with a pregnancy complicated by hyperglycaemia. The detailed methods of this intervention have been reported previously [25]. Formative work in 2016–2017, underpinned by the Systems Assessment Tool [26], identified five intervention components [16,17]:Increasing workforce capacity, skills and knowledge, and improving the health literacy of women;Improving access to healthcare through culturally and clinically appropriate pathways;Improving information management and communication;Enhancing policies and guidelines;Embedding clinical registers for women with hyperglycaemia in pregnancy—the NT and FNQ Diabetes in Pregnancy (DIP) Clinical Registers—within models of care.

Implementation activities were developed to address these components (Appendix A and Appendix A). These activities were implemented across all health service levels (primary, secondary and tertiary care) throughout three study regions (Central Australia and Top End in the NT, and FNQ) between 2016 and 2019.

### 2.3. Evaluation Methods

A mixed-methods evaluation of the health systems intervention was conducted, aiming to determine the impact of the intervention on systems of care and on birth outcomes. Detailed evaluation indicators were developed (Appendix A), aligning with the constructs of the RE-AIM framework (Reach, Effectiveness, Adoption, Implementation, Maintenance) [27]. Evaluation design and methods have been previously published [25]. The evaluation used data from multiple sources—qualitative interviews, a health professional survey, the NT and FNQ DIP Clinical Registers—and data from primary care electronic health records. For this paper, where we sought to explore health professional perspectives of systems of care after program implementation, we analysed interview and survey data. Interview and survey topics were informed by evaluation indicators (Appendix A). The methods for interviews and surveys are summarised here, with additional detail provided in Appendix A.

#### 2.3.1. Interviews

Methods for this qualitative component of the evaluation were underpinned by a phenomenological methodology. Interview data were collected from evaluation case study sites. These sites included six primary care services (one government and one Aboriginal Community Controlled Health Service in each of three study regions (Top End—Danila Dilba Health Service, Central Australia—Central Australia Aboriginal Congress, and FNQ—Apunipima Cape York Health Council)), in addition to the major referral hospital in each study region. Clinicians employed at case study sites whose role involved the care of women with hyperglycaemia in pregnancy were recruited to participate in qualitative semi-structured interviews. In addition to clinicians, health service managers and policy makers at the regional level and the study implementation team were invited to participate. Purposive sampling was used to ensure a diversity of representation across professions (e.g., nursing, midwifery, medical, etc), health service type (e.g., primary care, hospital) and setting (e.g., regional centre, remote) in each study region. Participants were recruited through email or telephone.

Author D.M., a clinician-researcher with five years of experience working in the NT, interviewed participants by phone, online or in person between March and October 2020, with most interviews completed by late May. Due to the COVID-19 pandemic and restrictions on both travel and face-to-face contact in health services, most interviews were conducted by phone or online (Zoom). Interview topics included practice changes, health service changes, perceived support, confidence and knowledge. Interviews were audio-recorded and transcribed verbatim by an external provider (Outscribe). All participants were offered the opportunity to review their transcript; 18 participants accepted this opportunity, with two participants submitting additional written notes after reviewing their transcript, which were included in data analysis. 

Author D.M. analysed qualitative data using a hybrid inductive–deductive approach. Coding for the deductive phase was underpinned by the pre-specified evaluation indicators (Appendix A). The coding structure was refined through discussion with authors N.F., an evaluation officer with extensive experience working in the remote Australian context, and R.K., a qualitative researcher with over 10 years of experience conducting research with Aboriginal communities. Themes were explored across the whole study and by study region. NVivo (version 12; QSR International) was used to assist with data management and analysis.

#### 2.3.2. Surveys

Health professionals across the three study regions were invited to participate in surveys regarding care provision to women with hyperglycaemia in pregnancy at two timepoints: pre-intervention/baseline (FNQ: November 2016–April 2017, Top End and Central Australia: May–November 2017) and post-intervention survey (all regions: February–April 2020). Any clinician working in the NT or FNQ providing care during and/or after pregnancy to women with hyperglycaemia in pregnancy at baseline and/or post-intervention was eligible to participate. As there is a high turnover of clinical staff across the study regions, participants were recruited independently at each timepoint. The development of survey questions was informed by issues identified during regional workshops with clinicians. Baseline survey topics in FNQ focussed on care during pregnancy, while in the NT survey topics included care during and after pregnancy. Post-intervention topics contained some modifications due to adaptations made during study implementation, including additional questions relating to postpartum health. Further detail regarding survey topics and methods of survey development and distribution are included in Appendix A. 

Descriptive statistics (frequencies and percentages) were reported for pre- and post-intervention, across the entire study and by study region. For survey items where data were available, both pre- and post-intervention, unadjusted comparisons were made using univariable logistic regression. Responses for each survey were not linked and participants were anonymous, thus analyses were unpaired. To adjust for possible confounding due to differences in participant characteristics between baseline and post-intervention surveys, multivariable logistic regressions were conducted. Independent variables with *p* value ≤ 0.2 on univariate analysis were included in the multivariable model building process. Among these variables, those with *p* value ≤ 0.1 on stepwise multivariable analysis were included in the final model for each outcome. For logistic regression, multinomial outcome variables were recategorised to binomial variables (categories aligned with best clinical practice according to local guidelines were codified as 1 and any other response was codified as 0). Analyses were performed with STATA 15.0 (StataCorp, College Station, TX, USA) with the threshold for statistical significance defined as *p* < 0.05. Free text responses were thematically coded, using NVivo V12.

#### 2.3.3. Ethics and Governance

This study was approved by the Central Australian Human Research Ethics Committee (approval HREC-15-345), the Human Research Ethics Committee of the NT Department of Health and Menzies School of Health Research (approvals HREC 2015-2461 and 2018-3189), and the FNQ Research Ethics Committee (HREC/16/QCH/15-1029). Evaluation participants provided written informed consent. This evaluation had formal approval from evaluation case study sites.

This study was informed by the Partnership’s Aboriginal and Torres Strait Islander Advisory Group, who were consulted regarding implementation priorities and activities, evaluation methods and interpretation of evaluation findings. Author S.Co. represents the Advisory Group in this manuscript. 

## 3. Findings

A total of 46 participants were interviewed (Table 1). Mean interview duration was 42 min (range 20–69 min), with the majority (83%) being longer than 30 min. Six interview participants publicly identify as Aboriginal or Torres Strait Islander people. 

The survey had 183 participants at baseline and 137 post-intervention (Table 2). In the post-intervention survey, 17% of respondents stated they had previously participated in a diabetes in pregnancy survey, although as participants were anonymous we were unable to calculate the proportion of health professionals who completed both the baseline and final evaluation survey, or to match participant responses across timepoints. Midwives represented a large proportion of respondents at both timepoints (31% baseline, 49% post-intervention). At both baseline and post-intervention, more than half of respondents provided care predominantly to Aboriginal and/or Torres Strait Islander clients (baseline 43%, post-intervention 53%), and more than 75% worked predominantly in primary healthcare.

Participating health professionals reported changes in individual clinician practice, health systems and health professional confidence and knowledge relating to care for women with hyperglycaemia in pregnancy. They also identified several opportunities for ongoing improvements.

### 3.1. Health Professional Clinical Practice

Most survey participants (64.1%) reported changing their practice in caring for women with hyperglycaemia in pregnancy during the health systems intervention (Table 3). Post-intervention, a greater proportion of clinicians reported screening for hyperglycaemia in the first trimester for women with risk factors for hyperglycaemia in pregnancy, as per guidelines [28,29] (baseline 68%, post-intervention 81%, *p* < 0.001). Post-intervention, clinicians were more likely to report using the 75 g oral glucose tolerance test most commonly for first-trimester screening (although the change was borderline significant on multivariable analysis: baseline 50%, post-intervention 74%, crude *p* < 0.003; multivariable adjusted *p* = 0.06). There was an increased recognition of the criteria for screening for hyperglycaemia in early pregnancy (Appendix A).

Interview findings supported the survey results regarding improvements to care during pregnancy. Participants reported that, following the health systems intervention, women were more likely to be referred for specialist management of hyperglycaemia in pregnancy, and referral was occurring at an earlier gestational age:

I think certainly some clinicians realise that as a midwife we’re a generalist, and that you really can’t care for women who have diabetes in pregnancy without specialised support services. (Midwife, primary care, FNQ)

So we’ve been seeing people earlier I think... I think raising awareness of that early screening has potentially made a change. (Medical practitioner, hospital, FNQ)

And usually it’s [referral] been quite close to diagnosis … whereas previously we might have picked them up quite late and not had the opportunity to see them … we’ve sometimes been able to have a bit more ongoing input throughout their pregnancy rather than just that little tail end. (Dietitian, primary care, Central Australia)

In addition to improvements in antenatal care, interview participants reported that the health systems intervention had resulted in increased attention being paid to postpartum follow-up, facilitated through the use of reminders, recalls and care plans:

So also just that, like one of the things that the project probably reminded me of more is also the follow-up. Like I’ve known the follow-up had to happen, but just coming up with different ways within an electronic system to try and make it happen. (Diabetes educator, primary care, FNQ)

So I think our postpartum screening might have improved, and I think a lot of that is around communication and reminders. (Midwife, hospital, FNQ)

Survey results also indicated improvements in postpartum diabetes screening practices, with clinicians more likely to report assessing glycaemic status at the recommended 6–12 weeks following pregnancy [28,29] for women with GDM (baseline 28%, post-intervention 67%, *p* < 0.001). Although the timing of postpartum glucose testing for women with pre-existing diabetes also improved, many clinicians still resumed HbA1c testing in women with pre-existing diabetes within 12 weeks postpartum, i.e., earlier than recommended (56% final survey; 78% baseline, *p* = 0.03 on multivariable analysis). 

### 3.2. Health Systems 

Interviewees identified multiple health system level changes in the care of women with hyperglycaemia in pregnancy. The health systems intervention was seen as being important in influencing these changes, particularly in the establishment of a network of supportive clinicians. It was reported that this led to the expertise in referral centres becoming more readily accessible to primary care clinicians:

We have regular conferences with the obs and gynae … I think there’s more of a team, we’re all on the same team, and you don’t have the ivory towers that you had before. (Diabetes educator, primary care, Top End)

They [primary care clinicians] know the people [at referral centre] and they know how to get in touch with them so that’s made a big difference. (Midwife, hospital, Central Australia)

The capacity to network, being part of that wider reference group. Especially when you’re working remote, you don’t get to network a lot with people across services. (Medical practitioner, primary care and hospital, Central Australia)

Interview findings regarding this network were supported by survey results. The majority of survey respondents (78%) felt well-supported in providing care for women with hyperglycaemia in pregnancy, with 58% reporting that support had improved during the health systems intervention. Many more survey participants (75.9% compared with 1.5% at baseline; Figure 1) reported seeking advice from an expert clinician in the final evaluation survey. 

One interviewee described how the ability of primary care clinicians to access increased support from hospital diabetes services had facilitated a more woman-centred approach to care, with the ability to offer women greater flexibility regarding where they attended for care during their pregnancy:

We also give the ladies a choice where they’d like to go. So at the start, when I first started… it seemed like they were wanting all the ladies to go to [hospital] clinic. But a lot of the ladies were leaving. It wasn’t, didn’t feel culturally appropriate. So now we say where would you like to go? You can come to [primary care clinic] or you can go to [hospital] or you can do a bit of both. (Diabetes educator, primary care, Central Australia)

There were some reports of improved communication between hospitals and primary care, although this was inconsistent:

The opportunities have opened up for us to be able to communicate in a better way, so that we all can work collaboratively in a group instead of in our own silos. (Diabetes educator, primary care, Central Australia)

I’ve had a lot of feedback from our rural sites that communication is so much better and the outcomes are much better and women are happier and things like that. (Midwife, hospital, FNQ)

Sometimes you get feedback from the antenatal clinic when they go down for appointments and stuff… sometimes it’s good, sometimes it’s completely absent, sometimes it’s timely and sometimes it’s three or four weeks later when you’ve asked for it …that’s a real struggle. (Midwife, primary care, FNQ)

The health systems intervention, including data from the NT and FNQ DIP Clinical Registers, was perceived as important in providing evidence to justify the allocation of resources to provide care for hyperglycaemia in pregnancy:

The Partnership’s provided the foundation, I suppose, so the foundation for me to be able to say yep, this is why we’re doing this [laughs] and this is why we’re investing in a clinic being set up this way. (Medical practitioner, hospital, Central Australia)

The DIP Clinical Register data were also seen as important in identifying areas for quality improvement:

What reports we had coming off the clinical register were showing that our breastfeeding rates [for women with type 2 diabetes] on discharge were reasonably low and that within 12 months were low as well and there’s been some discussions around that... And so it’s been a discussion about … what we’re recommending, but these are some of the reasons that women aren’t breastfeeding and those reasons need to be heard. (Implementer and Diabetes educator, hospital, Top End)

… maybe that approach can inform practice to say, you know, we need extra services for our high-risk Aboriginal population. (Diabetes educator, primary care, Central Australia)

These findings highlighted the utility of the data from the DIP Clinical Registers.

### 3.3. Health Professional Confidence and Knowledge

The proportion of survey respondents who were confident in managing women with hyperglycaemia in pregnancy increased post-intervention (baseline 60%, post-intervention 72%, *p* = 0.003) (Figure 2). Confidence in postpartum management also increased (very confident/confident at baseline 57%, post-intervention 72%, *p* = 0.01).

Interviewees reported that improvements in clinician confidence were attributable to the health systems intervention, through increasing clinician knowledge:

Definitely, they’re [clinicians] talking to them [women] because they’ve got more information on that understanding about the diabetes in pregnancy and it’s like anything, you have the information, it empowers you to empower them or to empower anybody. (Diabetes educator, primary care, Central Australia)

Interviewees also reported that the health systems intervention had increased awareness and interest in hyperglycaemia in pregnancy among both clinicians and women:

It did raise the profile of diabetes in pregnancy and the importance of it and how, actually, it’s everyone’s business. (Midwife, primary care, Central Australia)

With the information that we deliver with some of the brochures, women are wanting more knowledge, they are seeking more knowledge. (Midwife, primary care, Central Australia)

### 3.4. Opportunities for Further Improvements

Despite the progress made through the health systems intervention, participants identified several opportunities for further improvements in the care of women with hyperglycaemia in pregnancy. Survey respondents indicated a need for more culturally appropriate education and resources; this aligned with comments from interviewees:

It’s very obvious to me down on the ground that there’s a whole heap of cross-cultural communication stuff that has to happen better, and there’s totally different models of disease and illness that we haven’t really put diabetes into a good conversation with. (Medical officer, primary care and hospital, Central Australia)

I think some clinicians believe in the autonomy of the woman, and not that I don’t believe in the autonomy of the woman, I do believe in the autonomy of the woman, but I think sometimes the autonomy of the woman can only be created if she receives enough education to truly be able to have informed autonomy. (Midwife, primary care, FNQ)

As mentioned earlier, information-sharing between health services was still frequently identified as being problematic, despite improvements having been made:

The bottom line is that most of our clients already think we share records, they can’t believe that you don’t understand what happened in at [referral hospital] when they came out. (Diabetes educator, primary care, FNQ)

Interviewees also identified that action to address the impacts of the social determinants of health was essential to improve care and outcomes for women with hyperglycaemia in pregnancy:

In addition to food security issues, which are still ongoing, housing problems, which are still ongoing, domestic abuse, and other stuff. All of those play an important part into whether a woman is able to engage in the services. I think diabetes management would again be reflective of general socioeconomic progress in them, as they advance in that field, I think diabetes management will also become better. (Medical officer, hospital, Top End)

Additional opportunities identified by survey respondents included the following: increasing the workforce, including the Aboriginal and Torres Strait Islander workforce, for providing care for hyperglycaemia in pregnancy, particularly dietitians and diabetes educators; improving access to services for remote-dwelling women; and improving access to pre-pregnancy counselling.

## 4. Discussion

Clinicians reported multiple improvements in care for women with hyperglycaemia in pregnancy in Australia’s NT and FNQ following implementation of a complex health systems intervention. Among these improvements were a more woman-focussed model of care and improved screening and monitoring for hyperglycaemia during and after pregnancy. Data from the NT and FNQ DIP Clinical Registers were perceived as supporting investment in health services for women with hyperglycaemia in pregnancy. These impacts align with the desired short-term outcomes of the study as specified in the program logic model (Appendix A), namely improved communication and care coordination, improved cultural appropriateness, and increased clinician skills and confidence.

Clinicians described an enhanced, patient-centred model of care with the ability to offer women greater flexibility regarding shared care arrangements. The health systems intervention was seen as facilitating the development of this model through increased collaboration between hospital and primary care clinicians, enabling provision of high quality care through primary care services with support from external specialists. The option for Aboriginal and Torres Strait Islander women to choose to receive care through an Aboriginal Community Controlled Health Service, rather than being required to attend all clinic appointments at a hospital, has the potential to enhance the cultural safety of care. Cultural safety has been previously demonstrated as a crucial factor in improving birth outcomes for Aboriginal and Torres Strait Islander women and infants [30].

Effective cross-cultural communication is another element required to enhance the cultural safety of care [31]. Health professionals in our study expressed a need to further strengthen cross-cultural communication relating to hyperglycaemia in pregnancy. This aligns with findings from our recently published study where Aboriginal women and community members expressed a need to improve community understanding of diabetes, especially the intergenerational risks associated with hyperglycaemia in pregnancy [32]. A genuine understanding of the relationship between hyperglycaemia in pregnancy, subsequent health risks and how such risks can be mitigated is essential to empower women and communities to improve their health. 

Post-intervention, a greater proportion of clinicians reported screening for diabetes during and after pregnancy in line with local guidelines [28,29]. Many factors influencing clinician behaviour have been identified in previous literature [33]. Use of the Systems Assessment Tool [26] as an underpinning framework during our formative work enabled identification of key barriers to providing optimal care for hyperglycaemia in pregnancy specific to the study setting. This facilitated a study design addressing the most relevant factors, including disjointed communication between service providers and variable clinician knowledge and confidence. Clinician self-reported findings from our evaluation indicate that this has been a successful approach. Use of the Systems Assessment Tool distinguishes our study from others in the literature; Huang et al.’s recent systematic review and meta-analysis reported that few studies aiming to improve postpartum diabetes screening among women with previous gestational diabetes identified an underpinning theoretical or logic model or conducted a process evaluation, leading to difficulty identifying the elements of an intervention that contributed to its success [34].

The NT and FNQ DIP Clinical Registers were instrumental in supporting the investment of health service resources for hyperglycaemia in pregnancy. The prevalence of hyperglycaemia in pregnancy has increased substantially in the NT over the last 30 years [5]. Northern Australia also has a very high prevalence of youth-onset T2D—in Central Australia, it is among the highest reported in the world [35]—thus, it can be anticipated that the proportion of women with pre-existing T2D when becoming pregnant will be high for the foreseeable future. There is an ongoing need for the NT and FNQ DIP Clinical Registers to monitor the epidemiology and outcomes for women with hyperglycaemia in pregnancy to support health services and policymakers. Our findings reinforce those of an earlier evaluation which identified the importance of the NT DIP Clinical Register in raising clinician awareness of hyperglycaemia in pregnancy [36]. The reported benefit of the NT and FNQ DIP Clinical Registers contrasts with Boyle et al.’s evaluation of a National Gestational Diabetes Register (NGDR) in Australia [37]. Their evaluation found that the NGDR, which issued reminder letters to women with GDM, had no impact on postpartum diabetes screening, and recommended that future use of the NGDR be integrated with primary care practices. This contrast illustrates that, while clinical registers can contribute to improvements in patient care and outcomes, consideration needs to be given from the outset to clinical register implementation.

We found some improvements in the communication between health services; this was consistent with findings from our previous interim evaluation [38]. However, perceptions regarding communication differed between hospital-based and primary care clinicians, with some in primary care identifying that they still did not receive timely, clinically relevant information. Previous studies aiming to improve care for women with hyperglycaemia in pregnancy usually focussed on improving care within a single health service [13], rather than recognising that women frequently interact with multiple services during the antenatal and postpartum period, and the need to address communication issues between services [16]. In healthcare more generally, failures in communication have been identified as being a major contributor to medical errors, as well as leading to significant waste of healthcare resources [39,40]. A need for further improvements in the information-sharing between services in the NT and FNQ was identified in our study. There are ongoing developments in hospital-based electronic health records within the study regions, which presents an opportunity to improve communication.

Survey findings identified areas where clinical practice can be further improved. While the timing of postpartum glucose screening had improved post-intervention, most respondents resumed HbA1c testing among women with pre-existing diabetes within 12 weeks, earlier than recommended [41]. In this early postpartum period, HbA1c may be artificially low due to physiological changes of pregnancy, including haemodilution and increased red cell turnover [42]; an artificially low HbA1c could lead to false reassurance regarding glucose levels. Our results suggest further clinician education is still required regarding the rationale for delaying HbA1c testing until 12 weeks postpartum.

The use of mixed methods, incorporating surveys and interviews, was a strength of this study. Surveys enabled the collection of responses from a broad sample of clinicians while interviews allowed clinician perspectives to be explored in depth. Findings from these two data sources generally aligned. An additional strength was the purposive recruitment of six primary healthcare evaluation sites, ensuring representation across a diversity of services. Despite regional health system differences, evaluation findings across regions were also remarkably similar. This evaluation was further strengthened by the inclusion of Aboriginal and Torres Strait Islander perspectives through the input of the Partnership’s Advisory Group and Aboriginal and Torres Strait Islander investigators.

The component of our evaluation reported here is limited to the perspectives of health professionals. Clinical data from primary care services and data from the NT and FNQ DIP Clinical Registers will subsequently be analysed to determine if there is objective evidence of the changes in practice reported by clinicians. Further analysis of the data collected for this evaluation will explore the enablers and barriers that influenced the outcomes of this health systems intervention. Our evaluation is also limited by the absence of a control group. Inclusion of a control group was not feasible due to several intervention activities (e.g., embedding care plans in electronic health records) being implemented at a region-wide level, and thus separation of control and intervention sites was not possible. Additionally, study stakeholders expressed concerns regarding withholding or delaying implementation of intervention components to a control group, and this was not considered acceptable by these stakeholders. Other limitations regarding the survey included the potential for response bias and the inability to determine response rates due to the use of snowball sampling.

Limitations were placed on our data collection activities for this evaluation due to collection coinciding with the early response to the COVID-19 pandemic. Many participants were unable to be interviewed in person due to restrictions on travel and face-to-face contact. Dissemination of surveys was also restricted in FNQ during this period due to the prioritisation of health service resources and messaging to those related to COVID-19, leading to a lower than anticipated number of survey participants in FNQ. Nevertheless, to our knowledge, ours is the first study in the literature reporting the impacts of a health systems intervention to improve care during and after pregnancy for the high-risk population of Aboriginal and Torres Strait Islander women with hyperglycaemia in pregnancy.

## 5. Conclusions

Health professionals in Australia’s NT and FNQ reported multiple improvements to clinical practice and health systems in providing care during and after a pregnancy complicated by hyperglycaemia following the implementation of a multi-component health systems intervention. Areas requiring ongoing attention include the development of culturally appropriate educational resources and further improving communication between health services. 

## Figures and Tables

**Figure 1 ijerph-21-01139-f001:**
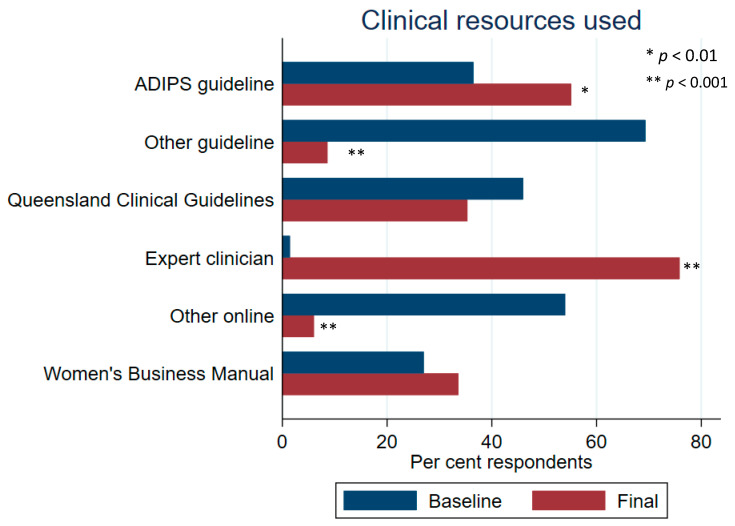
Resources used by clinicians for information on providing care to women with hyperglycaemia in pregnancy; ADIPS = Australasian Diabetes in Pregnancy Society; “other” resources listed by clinicians included the following: Queensland Statewide Clinical Network; South Australia perinatal guidelines; local hospital/health service policies; Royal Australian and New Zealand College of Obstetrics & Gynaecology guidelines; Royal Australian College of General Practitioner Guidelines. Baseline *n* = 137; post-intervention *n* = 119.

**Figure 2 ijerph-21-01139-f002:**
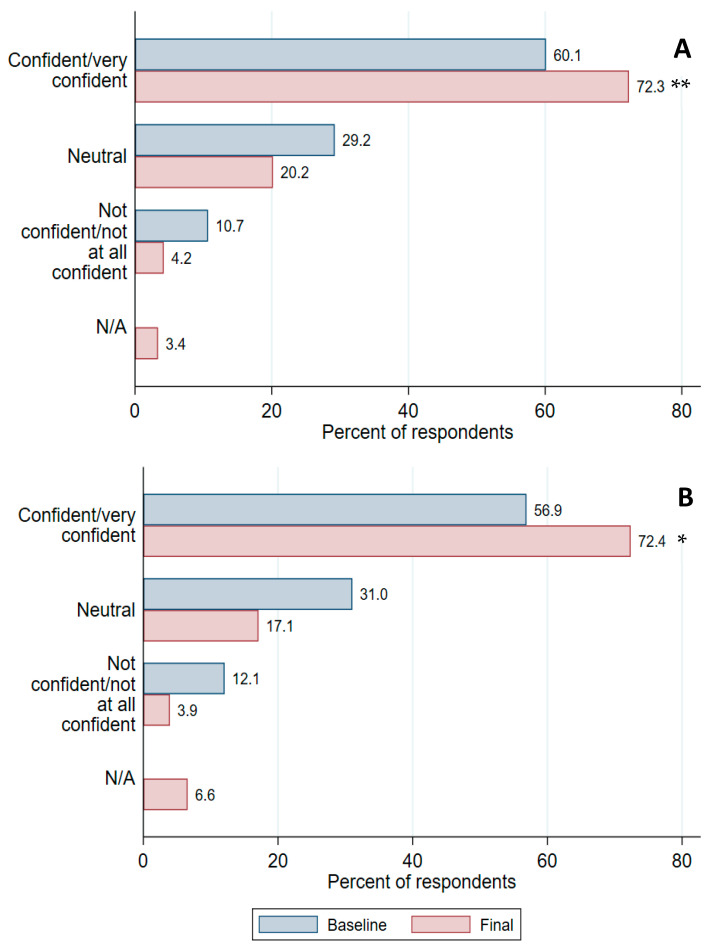
Health professional confidence in providing care during (**A**) and after (**B**) a pregnancy complicated by hyperglycaemia, before and after implementation of a multi-component health systems intervention; * *p* < 0.05; ** *p* < 0.01.

**Table 1 ijerph-21-01139-t001:** Interview Participants.

Profession	FNQ	CA	TE	Total
Midwife	6	6	3	15
Medical practitioner	2	4	5	11
Diabetes educator	3	3	2	8
Aboriginal health practitioner/Aboriginal health worker	1	0	2	3
Dietitian	0	2	0	2
Administrator	1	0	0	1
Implementer	2	1	3	6
Total	15	16	15	46

Total includes 24 participants interviewed by phone, 17 interviewed online (Zoom) and 5 interviewed in person; FNQ Far North Queensland; CA Central Australa; TE Top End.

**Table 2 ijerph-21-01139-t002:** Characteristics of Survey Participants.

Characteristic	Baseline N = 183 (%)	Post-Intervention N = 137 (%)
**Occupation**		
Nurse	9 (4.9)	5 (3.6)
Midwife	57 (31.1)	67 (48.9)
General Practitioner/GP Obstetrician	38 (20.8)	10 (7.3)
Medical Specialist	9 (4.9)	5 (3.6)
Medical Practitioner (other)	3 (1.6)	1 (0.7)
Diabetes Educator	23 (12.6)	25 (18.2)
Aboriginal and/or Torres Strait Islander Health Practitioner	12 (6.6)	3 (2.2)
Dietitian	12 (6.6)	11 (8.0)
Other *	17 (9.3)	10 (7.3)
Region		
Northern Territory—Central Australia	36 (19.8)	40 (29.2)
Northern Territory—Top End	45 (24.7)	52 (38.0)
Far North Queensland	101 (55.5)	45 (32.8)
Main work setting		
Regional/remote	117 (67.6)	73 (53.3)
Urban	56 (32.4)	64 (46.7)
Time in current position		
<1 year	33 (18.1)	15 (11.0)
1–5 years	68 (37.4)	54 (39.7)
5–10 years	32 (17.6)	38 (27.9)
>10 years	49 (26.9)	29 (21.3)
Client ethnicity		
Predominantly Aboriginal and/or Torres Strait Islander	76 (42.7)	72 (52.6)
Predominantly non-Indigenous	12 (6.7)	17 (12.4)
Mixed	90 (50.6)	48 (35.0)
Previous participation in Diabetes in Pregnancy survey	N/A	
Yes		23 (16.8)
No		75 (54.7)
Unsure		39 (28.5)
Primary place of work in primary health care		
Yes	143 (78.1)	104 (76.5)
No	40 (21.9)	32 (23.5)

* Other includes: Baseline—Aboriginal Health Practitioner/Diabetes Educator (*n* = 1), Health Promotion Officer (*n* = 2), Educator/Clinical Nurse Educator (*n* = 2), Indigenous Health Practitioner/Manager (*n* = 1), Medical receptionist (*n =* 1), Midwife/Child Health Nurse (*n =* 2), Registrar (Obstetrics and Gynaecology) (*n =* 1), Nurse/Midwife/Manager/Lactation Consultant (*n =* 1), Physiotherapist (*n =* 1), Registered Nurse/Midwife (*n =* 3), Registered Nurse/Midwife/Diabetes Educator (*n =* 2); Final—Nurse Practitioner (*n =* 2); Registered Nurse/Midwife (*n =* 4); Nurse/Midwife Educator (*n =* 1); Strong Woman Worker (*n =* 2).

**Table 3 ijerph-21-01139-t003:** Health professional survey results.

	Baseline *n* (%)	Final *n* (%)	*p* Univariable Logistic Regression	*p* Multivariable Model
Clinical practice				
Practitioners screening for diabetes in early pregnancy ^#^				
Yes	95 (67.9)	81 (81.0)	<0.001	<0.001 ^a^
No	32 (22.9)	1 (1.0)		
Unsure	13 (9.3)	1 (1.0)		
N/A—not viewed as part of practitioners’ role ^	N/A	17 (17.0)		
Screening test most commonly used in first trimester ^#^				
75 g oral glucose tolerance test	75 (50.0)	56 (73.7)	0.003	0.060 ^b^
HbA1c	33 (22.0)	16 (21.1)		
Random plasma glucose/blood glucose level	22 (14.7)	1 (1.3)		
Fasting plasma glucose/blood glucose level	3 (2.0)	1 (1.3)		
50 g glucose challenge test	3 (2.0)	2 (2.6)		
Unsure	14 (9.3)	0 (0.0)		
Screening test most commonly used in second/third trimester ^#^				
75 g oral glucose tolerance test	114 (83.3)	80 (95.2)	0.039	0.062 ^c^
HbA1c	9 (6.6)	1 (1.2)		
Random plasma glucose/blood glucose level	2 (1.5)	1 (1.2)		
Fasting plasma glucose/blood glucose level	1 (0.7)	0 (0.0)		
50 g glucose challenge test	7 (5.1)	2 (2.4)		
Unsure	3 (2.2)	0 (0.0)		
Gestational age of second/third trimester glucose screening				
<24 weeks	3 (2.2)	2 (2.3)	0.44	0.56 ^d^
24–28 weeks	128 (95.5)	81 (93.1)		
>28 weeks	2 (1.5)	2 (2.3)		
Other	1 (0.7)	2 (2.3)		
Proportion of women with hyperglycaemia in pregnancy seen postpartum for ongoing clinical care				
0–20%	54 (39.4)	33 (39.3)	0.11	0.074 ^e^
21–40%	21 (15.3)	7 (8.3)		
41–60%	18 (13.1)	8 (9.5)		
61–80%	19 (13.9)	13 (15.5)		
81–100%	25 (18.2)	23 (27.4)		
Practitioners screening for diabetes postpartum after gestational diabetes *~				
Yes	40 (80.0)	58 (77.3)	0.003	0.003 ^f^
No	6 (12.0)	1 (1.3)		
Unsure	4 (8.0)	0 (0.0)		
N/A—not viewed as part of practitioners’ role ^	N/A	16 (21.3)		
Use of recalls for postpartum screening *	44 (88.0)	57 (96.6)	0.43	0.43 ^f^
Timing of postpartum screening after gestational diabetes *				
Up to and including 6 weeks	25 (54.4)	13 (24.1)	<0.001	<0.001 ^f^
After 6 weeks, up to and including 12 weeks	13 (28.3)	36 (66.7)		
After 12 weeks, up to and including 6 months	6 (13.0)	3 (5.6)		
After 6 months, up to and including 12 months	2 (4.4)	2 (3.7)		
Screening test most commonly used postpartum after gestational diabetes *				
75 g oral glucose tolerance test	31 (64.6)	40 (74.1)	0.66	0.97 ^g^
HbA1c	12 (25.0)	11 (20.4)		
Random plasma glucose/blood glucose level	2 (4.2)	0 (0.0)		
Fasting plasma glucose/blood glucose level	2 (4.2)	0 (0.0)		
Unsure	1 (2.1)	3 (5.6)		
Resuming HbA1c monitoring postpartum in women with pre-existing diabetes *				
Up to and including 12 weeks	45 (81.8)	28 (56.0)	0.02	0.03 ^h^
After 12 weeks	10 (18.2)	22 (44.0)		
Made changes to own practice in caring for women with hyperglycaemia in pregnancy over the previous three years ^+^				
Yes	N/A	59 (64.1)	N/A	N/A
No		33 (35.9)		
Perceived support				
Perception of being well-supported in providing care for women with hyperglycaemia in pregnancy ^+^				
Yes	N/A	88 (77.9)	N/A	N/A
No		11 (9.7)		
Unsure		14 (12.4)		
Perception of change in support in providing care for women with hyperglycaemia in pregnancy ^+^				
Improved	N/A	65 (57.5)	N/A	N/A
Worsened		5 (4.4)		
No change		11 (9.7)		
Unsure		32 (28.3)		

N/A not applicable. Denominators for each item vary due to missing data: * Data for Northern Territory only as not asked in baseline survey in Far North Queensland; ^ N/A not a response option in baseline survey; ^#^ Excluded dietitian, diabetes educator, manager (occupations where >50% responded N/A in final survey); ~ Excluded dietitian, medical specialist (occupations where >50% responded N/A in final survey); ^+^ Asked only in final survey; ^a^ Adjusted for participant time in role and client ethnicity; ^b^ Adjusted for occupation, study region, time in role and primary place of work (primary health care or other); ^c^ Adjusted for occupation and client ethnicity; ^d^ Adjusted for time in role; ^e^ Adjusted for occupation and remotenes; ^f^ No covariables met criteria for inclusion in multivariable models; ^g^ Adjusted for occupation, study region and client ethnicity; ^h^ Adjusted for remoteness.

## Data Availability

Interview participants have not given consent for public sharing of their data, and this data may contain information which could identify participants or other individuals or organisations; as such, qualitative data will not be made available. Survey data are available from the corresponding author on reasonable request.

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
