# Peer review of "“It Empowers You to Empower Them”: Health Professional Perspectives of Care for Hyperglycaemia in Pregnancy Following a Multi-Component Health Systems Intervention"

_ijerph, 2024, doi:10.3390/ijerph21091139_

Round 1
Reviewer 1 Report
Comments and Suggestions for Authors
1. Is it safe to assume that this current manuscript is the analysis and results from the MacKay, et al ,2020 paper?
2. The design seems to be a non-cohort one-group pre-post intervention design. While many program evaluations use such a design, results may suffer from both internal and external invalidity.
3. To help with design limitations, the authors should consider adjusting for possible confounding variables across the baseline and post-intervention samples. Since I assume these are different samples, one could possibly use propensity score weighting to attempt a balance.
4. Why 183 participants at baseline and why 137 post-intervention samples were selected, what was the justification for these numbers?
5. It is too bad that 17% of respondents in both baseline and post-intervention could not be tracked or adjusted in the analysis.
6. I might suggest a consultation with a statistician to help with confounding issues.
References
MacKay D, Kirkham R, Freeman N, Murtha K, Van Dokkum P, Boyle J, et al. Improving systems of care during and after a pregnancy complicated by hyperglycaemia: A protocol for a complex health systems intervention. BMC Health Services Research. 2020;20(1):814.
Reviewer 2 Report
Comments and Suggestions for Authors
Firstly, I would like to congratulate the authors of the manuscript on their very interesting study on ““It empowers you to empower them”: Health professional perspectives of antenatal and postpartum
care for hyperglycaemia in pregnancy following implementation of a multi-component health systems intervention”. The study aimed at highlighting the health professional perspectives on the impact of the intervention on healthcare in The Northern Territory (NT) and Far North Queensland (FNQ) in Australia.
Authors highlighted health care professionals’ perspective for care of hyperglycaemia in pregnancy; however, it would be great if authors could include the other aspect from the target patients regarding their perspective for care for hyperglycaemia in pregnancy. This seems to be the major limitation of the study and authors should mention this in the discussion part. Authors should also try to further elaborate limitations and future research direction of the current study towards the end of manuscript.
In my opinion the manuscript should be accepted for publication with some minor observation that the authors should address before final submission.
1. Title seems too long; authors are requested to revise the title.
2. Address minor editorial issues throughout manuscript.
3. Define abbreviations at first place.
4. Minimize abbreviations and avoid abbreviations at the start of a sentence.
5. Discussion needs attention of the authors, please improve discussion section.
6. Authors should also try to further elaborate limitations and future research direction of the current study towards the end of manuscript.
Best of Luck.
Round 2
Reviewer 1 Report
Comments and Suggestions for Authors
No comments